# Primary Teeth Bite Marks Analysis on Various Materials: A Possible Tool in Children Health Risk Analysis and Safety Assessment

**DOI:** 10.3390/ijerph16132434

**Published:** 2019-07-09

**Authors:** Nikola Jovanovic, Bojan Petrovic, Sanja Kojic, Milica Sipovac, Dejan Markovic, Sofija Stefanovic, Goran Stojanovic

**Affiliations:** 1Department of Dentistry, Faculty of Medicine, University of Novi Sad, 21000 Novi Sad, Serbia; 2Faculty of Technical Sciences, University of Novi Sad, 21000 Novi Sad, Serbia; 3Laboratory for Bioarcheology, Faculty of Philosophy, University of Belgrade, 11000 Belgrade, Serbia; 4Department for Pediatric and Preventive Dentistry, School of Dental Medicine, University of Belgrade, 11000 Belgrade, Serbia; 5BioSense Institute, University of Novi Sad, 21000 Novi Sad, Serbia

**Keywords:** bite marks, exposure, primary teeth, health risk, toys

## Abstract

Background: All objects put into a child’s mouth could be hazardous in terms of trauma and toxic substance exposure. The aims of this study were to evaluate morphological characteristics of the primary teeth bite marks inflicted on various materials and to assess material wear using experimental model. Methods: Bite marks were analyzed on five materials: rubber, plastic, foil, wood, and silicone. In order to mimic children mouthing behavior an experimental setup has been designed using primary teeth placed in dentures and children’s equipment specimens. Results: Deciduous teeth make visible and recognizable traces when using physiological forces on all investigated materials. The most significant material loss was revealed in silicone samples, but it has been observed in all material groups, while mouthing with incisors using higher mastication forces were identified as significant predictors for material wear. There were no significant differences between type, species, and morphological-morphometric characteristics of the bite marks which are made by incisors, canines, and molars. Conclusions: In the range of physiological bite forces, deciduous teeth lead to wear of material from which toys are made while the analysis of bite marks in children equipment could give some information regarding the risk of trauma and exposure.

## 1. Introduction

It is commonly considered that exposure of children to various toxic and hazardous substances is reduced. Despite this, the risk of contact from childcare products and toys still persists from mouthing, chewing, and licking household stuffs, a habit that is widespread within the children of specific developmental phases. Understandably, child safety is of paramount interest. Keeping that in mind all child equipment and products should be as safe as possible, despite the fact that they are only being touched or put in the oral cavity. There are numerous approaches to health risk assessment in public and environmental health and new analytical tools are constantly employed in hazard assessment. The analysis of the pattern by which teeth can leave visible marks on different materials is, at the moment, at the intersection of various disciplines, including dentistry, public health, anthropology, archeology, and forensics.

Nowadays, great attention is paid to the safe use of toys since children come into contact with potentially toxic substances through oral mucosa and saliva [1]. Additionally, there is a danger of ingestion of small parts of toys during nibbling. In addition, despite the fact that dental enamel is the strongest tissue of the human body it is still certainly susceptible to fractures due to strong forces and repeated movements of the mandible with objects of various mechanical properties in the oral cavity [2,3]. The mouthing pattern of a child should be taken into consideration when evaluating overall exposure to various toxic and potentially dangerous substances present in children equipment since, during child development, mouthing together with observing, exploring, and touching present important parts of environment exploration [4]. Mouthing behavior has been categorized into three major types: licking, defined as object placement to the front of the oral cavity, sucking, where the object is placed completely into the oral cavity and, finally, chewing and biting, where the object is completely placed in the oral cavity and the child is chewing and biting on it [1,3,4].

Children younger than two years are at higher risk for exposure to greater concentrations of hazardous substances due to proven more frequent mouthing in combination with lower body mass, and this is the reason why it is important to anticipate not only the accurate estimation of the number of mouthing events during the day, but the actual effect of mouthing on mouthed objects, for example, the damage of the material surface, material wear, and possibilities of leaving visible bite marks and materials damage when using physiological masticatory forces during mouthing. Observational studies report that children up to four years of age may mouth specific objects up to 50 events during one hour. Moreover, a child wellbeing issue related to the safety of objects being put into the oral cavity of, particularly, younger children, is always present since some products are designed and intended to be put into the mouth, while, at the same time, some easily accessible products not intended to be chewed regularly come to pass in children’s mouths. Bite mark analysis is defined as the process of “the detection, recognition, description and comparison of bite marks that occur on living and inanimate objects caused by humans and animals” [5].

Some government institutions addressing the problem of environmental health and safety recently published reports and recommendations on plastic toys and equipment intended for use in children younger than three years, stating that children in this age group represent a distinct population that is highly vulnerable [6].

The aims of this study were to determine the morphological characteristics of the primary teeth-induced marks on the materials used for children equipment fabrication, using an experimental setup with controlled masticatory parameters and jaw models with deciduous teeth. Furthermore, the aim was to assess the extent of material wear using physiological masticatory forces.

## 2. Materials and Methods

This study analyzed primary teeth bite marks inflicted on 5 different materials: rubber (R), plastic (P), aluminum foil (A), wood (W), and silicone (S).

### 2.1. Preparation of Primary Dentition Models

In order to mimic child mouthing behavior an experimental setup has been designed using natural donor intact primary teeth placed in partial and complete dentures and children’s equipment specimens, as shown in Figure 1. The study protocol was approved by the Ethical Committee of the Dental Clinic of Vojvodina, approved 15 August 2017, protocol no. 127/17. All teeth were collected with the informed consent of the patient and written consent signed by the parents or caregivers. Experimental teeth were placed in two pairs of total dental prostheses, and the chewing simulation was performed in the non-Arcon type semi-adjustable articulator. The Bennett angle was set at 15°. Stone cast models of a child aged three years (actual patient at our department) with bite registration were used as a template for a set of donor primary teeth and removable denture fabrication. The procedure of denture fabrication followed the principles of conventional prosthodontic technique but with a few modifications. Models with bite registration and occlusal vertical dimension records were transferred into the abovementioned articulator.

The teeth on the study models were cut off and shellac base plates were adapted on the upper and lower casts and wax occlusal rims were then prepared. Since a face bow record was not used the interocclusal records were performed as follows: Casts in position of central occlusion were transferred into articulator and temporarily fixated using plasteline. The incisal pin was in contact with mesial surfaces of the mandibular central incisors, forming the 10.4 cm equisame triangle with the condyles. Then the upper cast was plastered and, after that, the lower cast was plastered. Then the teeth were cut and ahelac basis plates with wax were designed first for the upper and then for the lower teeth. One pair of complete dentures and one pair or partial dentures were fabricated using the same model, providing reproducibility for both pairs of dentures. Teeth were arranged following the basic priciples of intercuspation and characteristics of dentition. 

In addition, six single representative teeth (maxillary incisor, I; maxillary canine, C; mandibular second molar, M2; maxillary second molar, M1; maxillary first molar, M3; and mandibular first molar, M4) were mounted into acrylic blocks 1 cm × 1 cm × 1 cm and chewing simulation was performed using direct hand pressure.

The primary teeth of typical morphology were used in the experiment. When it comes to central maxillary incisors, teeth with a completely straight incisal edge and prominent cingulum were chosen. Lateral maxillary incisors with similar morphological characteristics as central, but proportionately smaller dimensions and rounded distoincisal edge were selected. All maxillary canines were with well developed, sharp, and pronounced cusps. The maxillary first molars all had three cusps, two buccal and one lingual, and a prominent mesiobuccal cervical bulge. The maxillary second molars had four defined cusps. The mandibular central incisors were symmetrically flat when viewed from the buccal with cingulum present on the lingual surface. The mandibular lateral incisors had similar morphological features, with incisal edges sloping distally, and distoincisal edge being more rounded. The maxillary canines had a pronounced cusp with the crown being shorter and narrower labiolingually. The maxillary first molars had four cusps, pronounced mesiobuccal bulge, and transversal ridge. The second maxillary molars had five cusps.

### 2.2. Preparation of Materials

Samples of all materials were cut into approximately the same dimensions (10 mm × 20 mm × 2 mm). For cutting, diamond cutting discs or scalpels were used. Five samples were made of each material: (a) wood samples from the wooden toys with the label CE; (b) silicone samples by cutting silicone pacifiers bought in a local pharmacy from different commercial producers; (c) rubber samples by cutting rubber toys with the label CE; (e) plastic samples by cutting three plastic toys; and (f) tin foil samples as the representative of the metal with which the child would come into contact was obtained by folding the film of the foil for the packing of children's chocolates bought at the consumer goods store and multiple (four times) bending along the longitudinal axis, as presented in Figure 2.

### 2.3. Mouthing Simulation

Five volunteers were included in the experiment under supervision of B.P. Two of the participants were fifth year students of dentistry, and three were working as specialists in pedodontics. Participants were directed how to simulate chewing and they were all familiar with the aim of the experiment. Initial calibration regarding the applied force has been performed using a digital scale and a scale with weights, and the intensity level was defined and agreed by consensus. The simulation of chewing using the articulator and direct hand pressure was performed using the scales with three defined levels of 1 kg, 10 kg, and 20 kg. In order to complement the experiment observations, all participants were asked to write down observations during the chewing simulation to identify the teeth that were predominantly or exclusively used for chewing simulation, the mechanism of mark formation, and intensity level.

### 2.4. Bite Mark Analysis

All samples were photographed from both sides using an electronic microscope (TM3030, HITACHI, Tokyo 105-8717, Japan—courtesy of the BioSense Institute, University of Novi Sad). with 20–50× magnification and a digital single lens reflex (DSLR) camera and macro objective (focal length 105 mm) (Nikon, Shinagawa, Tokyo, Japan).

For each individual mark obtained during the experiment the following parameters were recorded:(A)Referent image: Teeth mark identification was accomplished by comparison of the testing material sample images before and after the experiment. Specimens were photographed using SEM (TM3030, HITACHI, Tokyo 105-8717, Japan) or a Zeiss stereomicroscope (Zeiss, Jena, Germany). SEM evaluation was conducted using TOPO mode with uniform magnification at 40×, while observations on Zeiss stereomicroscope were performed using various magnifications from 20× to 50×. Data regarding the magnification, scale bar, and mode were available for each referent image.(B)Tooth type: Depending on a tooth predominantly or exclusively used in mark formation the tooth designation was attributed to each mark.(C)Intensity level: The force is classified as weak if it was up to 10 N, moderate if it ranged from 10 to 100 N, and strong if it was in the range of 100 N to 200 N.(D)Mark dimensions: The length, the width, and the relationship between the length and the width (L/B ratio) of each mark were recorded. Raw referent images obtained from the microscopes were processed and all measurements performed in Gwyddion open source software (Czech Metrology Institute, Jihlava, Czech Republic) as shown in Figure 3.(E)Mark type: All marks having lengths less than four times their widths were described as pits. The pit shape was classified into five groups: (1) round; (2) elliptical; (3) crescent; (4) drop-like; and (5) other. Scores were defined as marks with lengths four times their widths and longer. The typical appearance of various mark types has been shown in Figure 4 and Figure 5.

### 2.5. Statistical Analysis

In summary of the data obtained in this experimental study, nominal and categorical variables were presented as number and percent, while continuous variables were presented as the mean with standard deviation. Chi square test and one-way ANOVA, a collection of statistical models used to analyze variation among bite marks in five different materials with the post hoc Tukey HSD test for intergroup differences evaluation were used for each two groups’ comparison. Finally, binary logistic regression analysis with the material loss as the dependent variable was constructed. For statistical analysis, the open source statistical program Jamovi Project (2018), Jamovi (Version 0.9.2.8) retrieved from https://www.jamovi.org, developed by Jonathon Love, Damian Dropmann, Ravi Selker, was used with significance level set at 0.05.

## 3. Results

During this experimental study a total of 406 bite marks were recorded: 122 on plastic, 45 on silicone, 48 on rubber, 78 on wood, and 113 on thin foil. The sample size power calculation was performed using GPower 3.1 software (Heinrich Heine University Düsseldorf, Düsseldorf, Germany) and indicated that a sample size of N = 200 bite marks achieves 80% power (α = 0.20) to detect a difference in the effectiveness for the groups with a significance level of 0.05. A total of 406 measurements, which is an approximate twofold sufficient sample size, could be considered as representative for the investigated variables. 

The dimensions of bite marks ranged from 39 µm up to 3.8 mm. Average dimensions and all parameters of descriptive statistics of the bite marks on different materials are shown in Table 1 and Figure 6. 

When it comes to the results regarding mark length, one way ANOVA analysis revealed that the differences between five materials was not statistically significant (*p* = 0.9608). The Tukey HSD post-hoc test was additionally performed and revealed no significant differences between each pair of tested groups (*p* > 0.05). Similarly, one way ANOVA analysis of mark width showed no significant differences between all tested materials (*p* = 0.6160), and the Tukey HSD post-hoc test showed no significant differences between each pair (*p* > 0.05). In contrast to that, the results regarding the ratio between the length and width ratio of the marks (L/W) showed significant differences between tested materials (*p* = 0.0000), while the Tukey HSD post-hoc test revealed that the values for the L/W ratio in silicon samples were significantly higher compared to each out of four remaining tested materials, while between all other pairs of materials the difference was not statistically significant (*p* > 0.05).

Regarding the specific type distribution of bite marks analysis revealed that both types of bite marks, pits and scores were unequally distributed in the five material groups (*p* = 0.008479) and the results are shown in Table 2.

In contrast to this, the frequency of distribution of various shapes of the pits was not statistically significant (*p* > 0.05). Similarly, the majority of the recorded linear marks, scores, in all investigated materials were multiple, non-branched, straight and curved relatively uniformly distributed, without statistically significant differences between five tested materials. It has been observed that all bite mark types can be produced by each tooth type and no statistical differences were observed when analyzing nominal variable (tooth type) and distribution of pits and scores in all tested materials (*p* > 0.05). Loss, or visible wear of material was recorded in all investigated materials and the results are shown in Table 2, Table 3, and Figure 7.

Initially, material loss was analyzed with the respect to the material type and statistically significant differences were observed (*p* < 0.05). The material loss was additionally analyzed against all investigated nominal variables (material, tooth type, mark type, mark shape), ordinal variable (intensity level), and continuous variables (mark length, mark width, and L/B ratio) using binomial logistic regression test and the results are presented in Table 3. The binary logistic regression model was defined as follows: material loss was the dependent variable and designated as N, no material loss, and Y, material loss present. Initial correlation analysis of all predictors revealed that there is no significant correlation between predictors (r > 0.7, Pearson coefficient), and all predictors were included in the regression model. However, the Hosmer–Lemeshow test revealed the level of significance of *p* < 0.05 suggesting that the model is poor fit, requiring exclusion of some predictors. When predictors’ mark length L and mark width B were excluded and the derived variable L/B left in the model the Hosmer–Lemeshow test revealed the level of significance of *p* > 0.05, suggesting a good fit.

As shown in Table 3 two predictors exhibited a significant contribution to material loss: Intensity level and tooth type (*p* < 0.05, values marked with asterisks). Bite marks inflicted using a high intensity level had 3.26 (OR = 3.261) higher odds for material loss compared to low and medium bite force.

## 4. Discussion

It is a well-known fact that all the objects put into a child’s mouth could be potentially hazardous in terms of mechanical trauma, with the risks of suffocation or choking being the most apparent and serious. In addition, mouthing of various objects also carries the risks of any object becoming stuck in the oral cavity and pharynx, together with foreign body incidents during which a baby can swallow an object. According to the available literature and the authors' best knowledge, this is the first study about the experimental analysis of bite marks produced by deciduous teeth on various materials designed for children equipment using the described methodology. This experimental study required a multidisciplinary approach, and carried with it both the risk and need for the merging of substantially humanistic, medical, and technical sciences. In the present investigation bite marks were analyzed in five different materials that are very common in households and intended for use in children.

Children at younger ages, during rapid development go through significant physiological changes that modify possible toxic and hazardous effects of various chemicals which renders them relatively more exposed in comparison to older children and adults, particularly when taking into account the actual concentrations of the chemicals and body mass [6]. It has been clearly described that, apart from chewing, there are numerous physiological movements during which different materials come into contact with the deciduous teeth, not with the primary purpose of swallowing the object or drinking it. When it comes to analysis of the type of the objects that children most frequently put in their mouth, analyses showed that children place a wide variety of items in their oral cavity, but it has been reported that plastic objects are the most commonly mouthed [6]. In this way, children can make visible traces on the surface of the object which is exposed to this para-masticatory functions. During intense and sudden mastication activities involving objects of more pronounced hardness there is a risk of teeth injuries. Additionally, there is a possibility of ingestion or aspiration small parts of the object. If the object that child puts into the mouth contains toxic substances, there is a risk of chronic exposure to toxic elements.

Data about the primary teeth bite marks in the area of forensic expertise are very scarce. Testing the possibilities of clarification of the way in which human primary teeth may leave detectable tooth marks on various materials demands a thorough examination of the interfering subjects, specifically the primary teeth enamel and various materials’ structures, together with the forces children are capable of generating during mouthing behavior. It has been reported that under laboratory conditions it is possible to generate the occlusion force up to 750 N on the molars and 250 N on the incisors, but it has also been suggested that mastication forces generated during regular chewing are significantly smaller, age and foodstuff related. Full rotary jaw movements with stable occlusal contacts start from about three years of age and from that period, child and adult oral motor functions are similar [7,8,9,10,11]. 

Linear shallow marks showing varying dimensions, width, and depth are consistently associated with the incisors biting action. Fernández-Jalvo and Andrews [12,13] provided the dimensions of these chewing marks for permanent teeth, ranging between 500 microns and 1.8 mm, and these authors recently suggested that these linear marks could be still narrower. In the present investigation it has been confirmed that the range of tooth mark sizes made by humans, and particularly children, is much greater. The results of the present investigation indicated that the examined materials have the ability to undergo very distinct and recognizable changes when subjected to children’s mouthing. According to numerous reports in RAPEX (Rapid Alert System for non-food dangerous products), there are abundant investigations regarding the presence of phthalates in the equipment for children and toys [14]. It has been clearly described that exposure may well occur from hand-to-mouth contact with these objects [15]. As a result, it is essential to decrease the possibility of contact of children with these products containing hazardous substances, as well to assess the amount, extent, and risk of exposure during all everyday activities of a child. In spite of this, a rather small number of studies have assessed the mouthing pattern in children younger than five years in depth [4]. It is well known fact that some components of toys and children equipment are constant source of critique and worry, as both parents and health professionals are of the opinion they may affect the health of the children. The majority of the concerns is directed to the presence of certain plasticizers, and most of the data available in recent studies focus on the presence of specific phthalates present in the plasticizers, but it has been emphasized that phthalic acid esters used in plasticizers are not the only concern [16]. 

The bite mark analysis on material other than bone and skin is not a new approach, but it is rarely used even in forensics [14,17,18,19]. For better understanding of the mechanism of bite mark creation, it is necessary to take into account the complex chewing mechanism and all temporomandibular joint movements. The bite force depends on factors which are individual for each person: masticatory muscles’ activity and occlusion [20]. A direct relationship between the number of mouthing events and age was observed, suggesting that children, to a lesser extent, put various items into their mouths as they grow up and two different groups, with respect to mouthing frequency, were established: in children younger than two years, an average of 80 mouthing events per one hour was noted, while in children older than two years these values decreased to around 40 episodes [4]. For the purposes of the present investigation direct observational approach was ruled out due to ethical and professional reasons, and analysis of the material itself without the control of teeth- and mastication-related parameters was considered less informative. It has been reported that the average maximum bite force at the primary teeth was in the range between 137 N and 361 N [21]. During the experiment in the present investigation forces up to 200 N were used.

Some emerging technologies were recently employed in bite mark analysis and bite marks were also analyzed in food, and there are observations and suggestions from these reports that 3D scanning should be additionally used in bite marks analysis, particularly in bite mark interpretation in highly deformable structures, such as food or skin, in order to record individual characteristic of the tooth structures [22]. 

Worldwide, there are regulations and necessary requirements for products intended for use in younger children, and corresponding to these regulations “childcare articles are products intended to ensure or facilitate seating, bathing, sleeping, transportation, and the movement and physical protection of children under four years of age“ [6]. In the framework of many studies addressing the issue of health and safety all these products are referred to as children’s equipment. In the present investigation, evaluation addresses the issue of potential exposure of younger children to five different substances commonly present in toys and children equipment when these objects are placed into the oral cavity and mouthed using physiological masticatory forces. Oral route exposure has been identified as one of the first assumptions when analyzing health risk assessments, since contact with hazardous substances can occur through saliva together with direct swallowing of particles as a consequence of chewing, and through these two completely different ways substances can be released either in the oral cavity through the processes of diffusion, through the mechanism of dissolution by gastric acid [23,24].

This study has been conducted in collaboration of three different scientific groups, namely pediatric dentists, physical anthropologists, and technical scientists. In our opinion, the present investigation has important implications in all these disciplines. From the perspective of pediatric dentistry, the public health issue with the use of children equipment and mouthing behavior is very important since the results obtained in this study could be used in the interpretation of the exposure risk and toxicity of the materials that are present in everyday children surroundings. In addition, parents who observe more intensive mouthing behavior in their children could pay additional attention regarding the availability of potentially toxic materials. Finally, the presence of recognizable, specific bite marks, with very distinguishable size and shape could be an indicator of excessive mouthing behavior.

## 5. Conclusions

Primary teeth leave visible and recognizable traces, when using physiological bite forces, on all tested materials. There are no significant differences between the type, species, and morphological and morphometric characteristics of the traces that are left by incisors, canines, and molars. In the range of the physiological forces, deciduous teeth lead to the wear of materials intended for use in children. Analysis of bite marks present on the materials designed for children equipment could give some important information regarding the risk of trauma and exposure of different hazardous materials.

## Figures and Tables

**Figure 1 ijerph-16-02434-f001:**
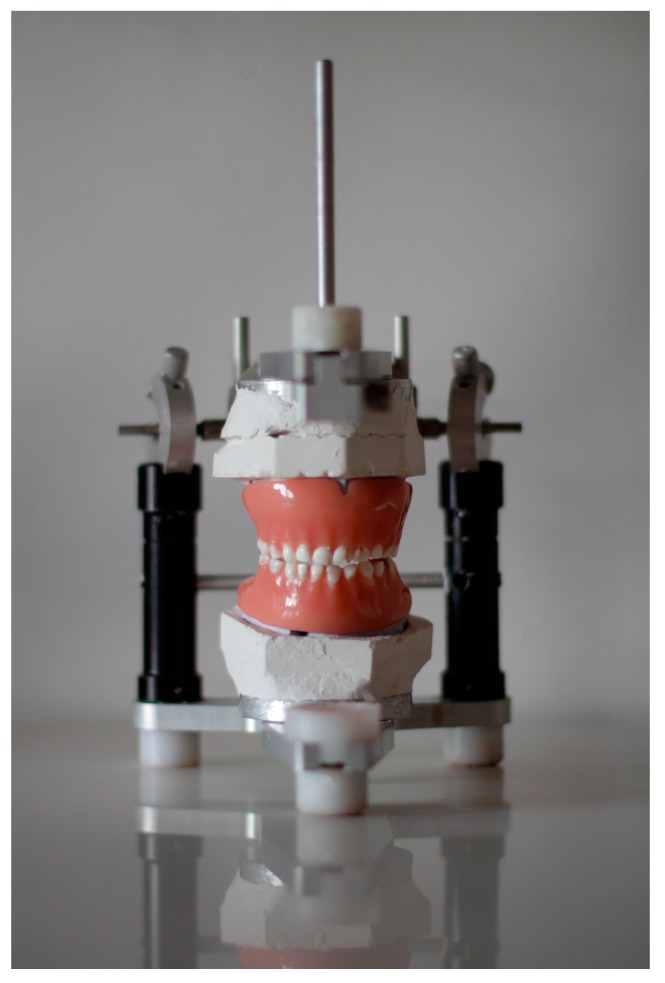
Experimental setup with primary teeth placed in the fabricated removable dentures and positioned into the articulator.

**Figure 2 ijerph-16-02434-f002:**
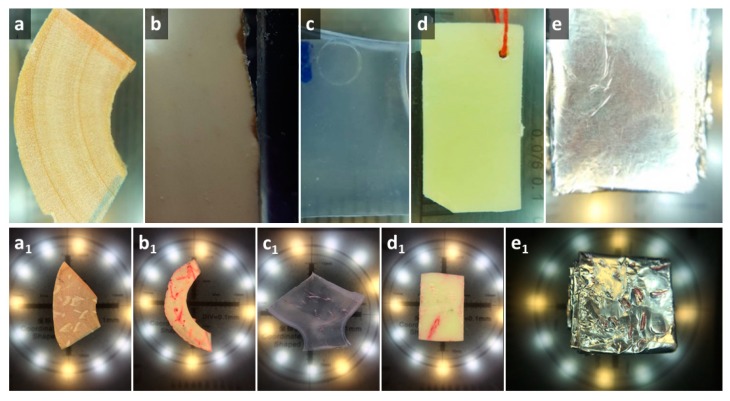
Tested materials photographed before and after the experiment: (**a**)–(**a_1_**) wood; (**b**)–(**b_1_**) rubber; (**c**)–(**c_1_**) silicone; (**d**)–(**d_1_**) plastic; and (**e**)–(**e_1_**) thin foil.

**Figure 3 ijerph-16-02434-f003:**
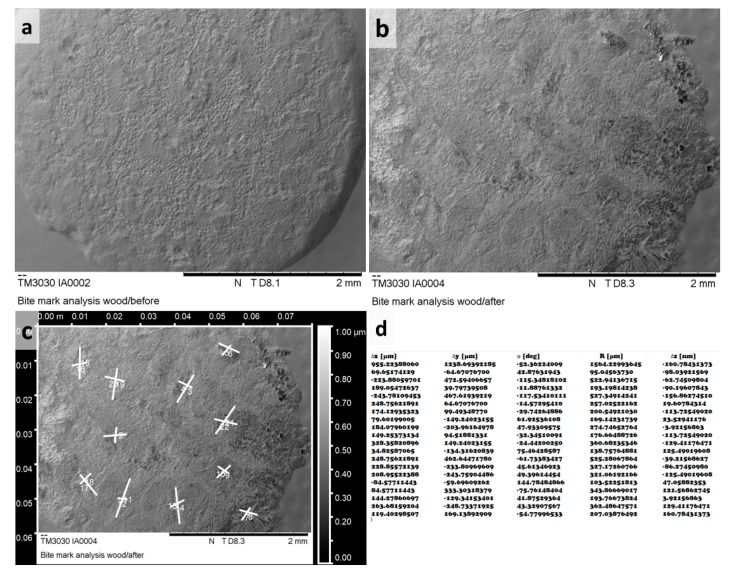
(**a**) SEM image of wooden sample before the experiment; (**b**) SEM image of the same sample after experimental biting; (**c**) bite mark designation and measurement using Gwyddeon software; and (**d**) actual measures for induced bite marks.

**Figure 4 ijerph-16-02434-f004:**
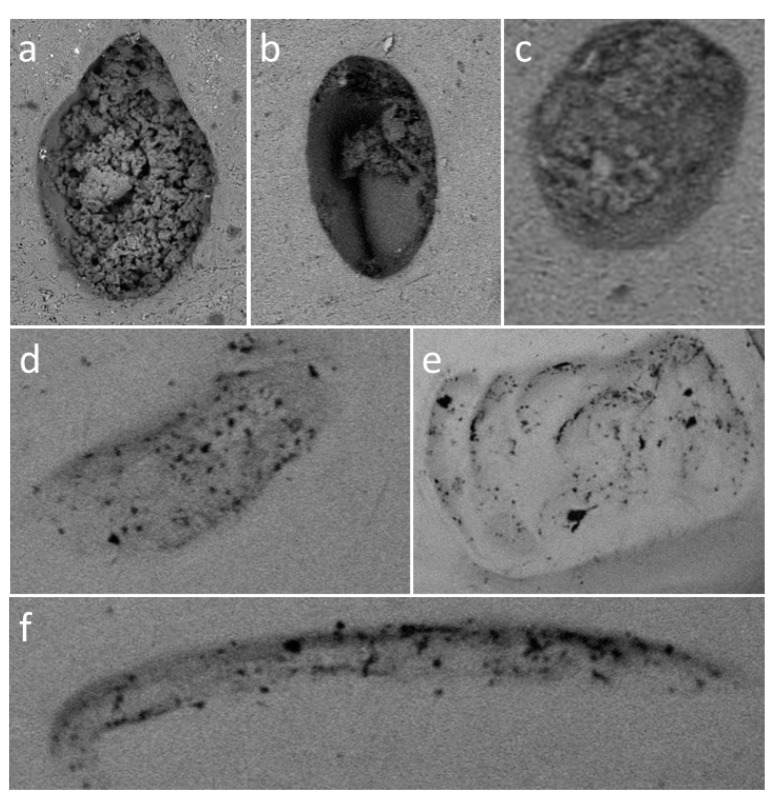
Bite mark types and shapes: (**a**) Drop-like; (**b**) elliptic; (**c**) round; (**d**) crescent; (**e**) other; and (**f**) score.

**Figure 5 ijerph-16-02434-f005:**
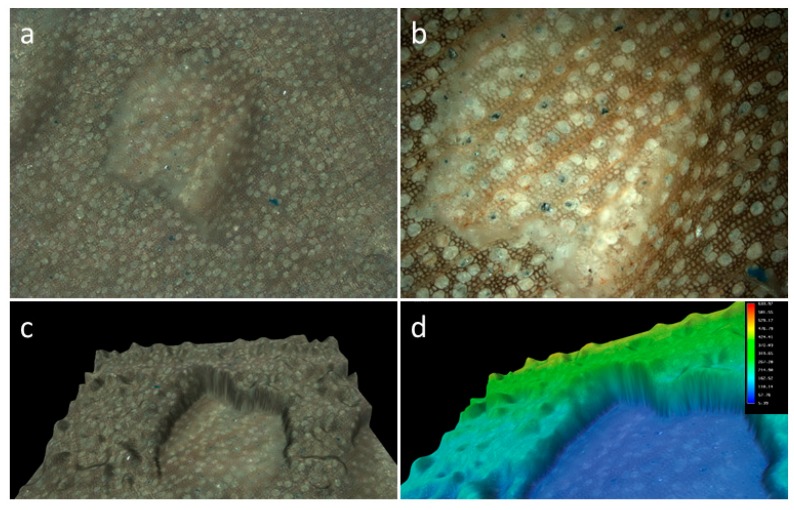
Bite marks obtained by optical profilometry on a wooden sample: (**a**) 2D at magnification 5×; (**b**) 2D at magnification 10×; (**c**) 3D profile; and (**d**) 3D profile in color scale.

**Figure 6 ijerph-16-02434-f006:**
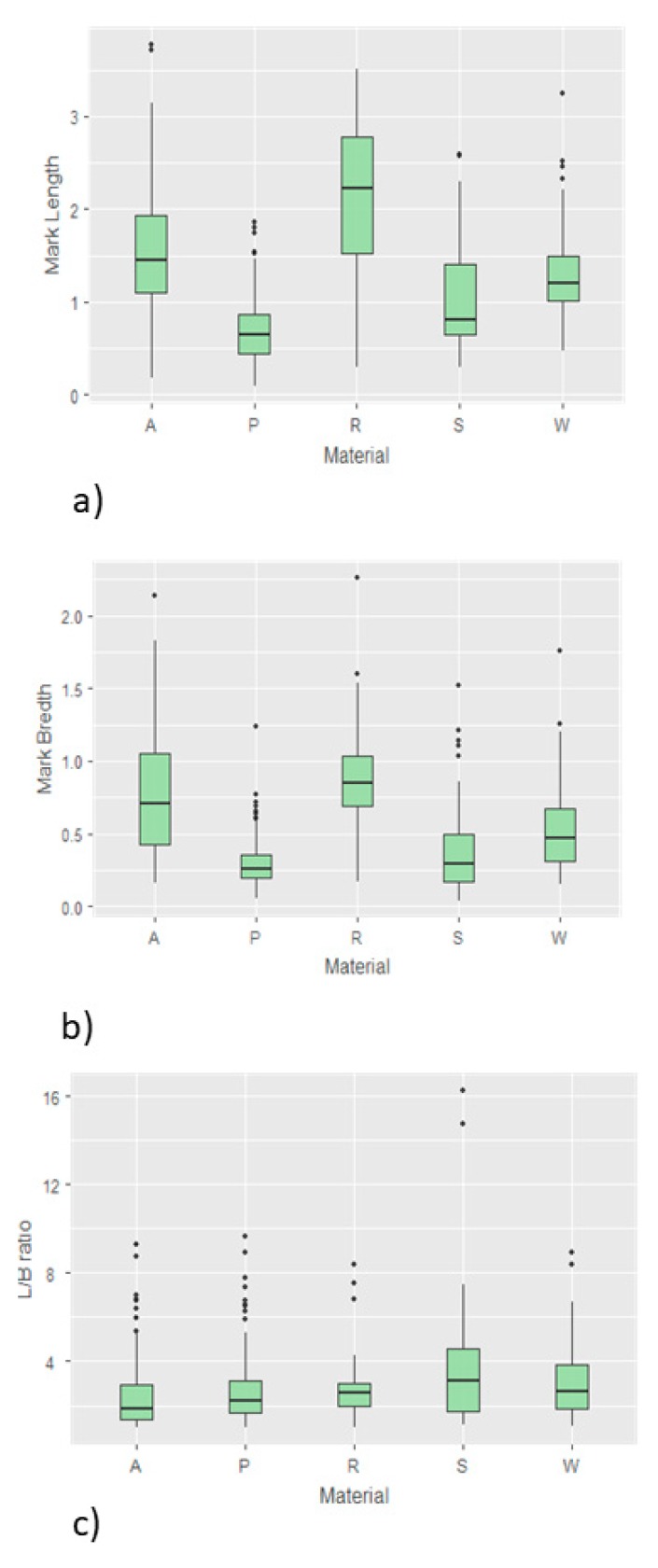
(**a**) Mark length, (**b**) mark width, and (**c**) L/B ratio.

**Figure 7 ijerph-16-02434-f007:**
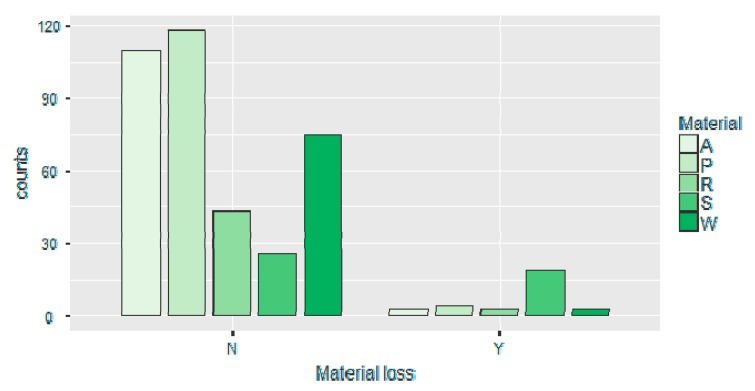
Material loss in relation to material type.

**Table 1 ijerph-16-02434-t001:** Descriptive statistics for the entire sample regarding morphometric parameters of bite marks. (mark length, mark breadth, and L/B ratio expressed in mm with mean, standard deviation (SD), and minimal and maximum values).

Material	No.	Mark Length				Mark Breadth				L/B Ratio	
		Mean	St. Dev.	Min.	Max.	Mean	St. Dev.	Min.	Max.	Mean	St. Dev.
Plastic	122	0.7	0.36	0.093	1.9	0.3	0.16	0.058	1.2	2.7	1.6
Rubber	48	2.2	0.75	0.29	3.5	0.88	0.37	0.18	2.3	2.8	1.5
Silicon	45	1.1	0.63	0.3	2.6	0.41	0.34	0.039	1.5	3.8	3
Al foil	113	1.6	0.69	0.19	3.8	0.77	0.42	0.16	2.1	2.6	1.8
Wood	78	1.3	0.49	0.48	3.2	0.53	0.29	0.16	1.8	3	1.6

**Table 2 ijerph-16-02434-t002:** Mark type and the material loss within five experimental material groups.

Material	Mark Type			Material Loss			
	No.	Pits %	Scores %	Yes	%	No	%
Plastic	122	86.89	13.11	2	1.64	120	98.36
Rubber	48	83.33	16.67	2	4.17	43	95.83
Silicon	45	68.89	31.11	19	42.22	26	57.78
Al foil	113	84.07	15.93	7	6.19	106	93.81
Wood	78	79.49	20.51	12	15.38	66	84.62

**Table 3 ijerph-16-02434-t003:** Logistic regression model for material loss prediction.

		*P*	OR	95% CI	
				Lower	Upper
L/B ratio		0.096	1.141	0.977	1.332
Intensity Level:	L + M	0.033 *	1.00 ^a^		
	H		3.261	1.101	9.654
Tooth type:	C	0.023 *	1.00 ^a^		
	I		10.478	1.386	79.193
	M	0.334	2.920	0.333	25.612

* *p* < 0.05; ^a^ odds ratio.

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
