# Peer review of "Primary Teeth Bite Marks Analysis on Various Materials: A Possible Tool in Children Health Risk Analysis and Safety Assessment"

_ijerph, 2019, doi:10.3390/ijerph16132434_

Round 1

Reviewer 1 Report

Abstract and introduction : Revise the abstract, the objective of the study is not clear.

Materials and methods: The study design is unclear. Once you mentioned partial dentures, once complete dentures. Two paragraphs are repeated (r. 61 and r. 69).

how did you get natural primary intact teeth? how many? how roots look like? were the resorbed? how did you find the right position in articulator? how did you set up the long axis?

what type of articulator has been used? was the articulator semi-adjustable or full-adjustable?

how did you simulate masticatory forces?

how did you follow the Bonvil triangle? how did you get the same parameters in two pairs of dental prosthesis?

how did you compare the hand pressure and articulator

there is no detailed specification of material used for the experiment, silicon vs. plastic

the intensity of force is unrealistic (200 N) in primary dentition

is the sample size representative?

Conclusion: What is the practical outcome of this study?

What is the novel finding of this study?

The study design is unclear and probably incorrect, not respecting basic gnathology principles. based on this the result are unrealistic.

Reviewer 2 Report

This paper discusses the primary teeth bite markers analysis on various materials using mimic primary dentition model. I gave some comments about this report. 

Major comments:

Introduction

The introduction is simply focused on the children’s problem, mouthing behavior with the toy. I easily understood the aim of this study.

Material and Methods

・The writing style is not structural, and hard to read. The authors should write more methodically like following; 1) preparing primary dentition model 2) preparing materials 3) the measurement procedure 4) statistic analysis.

・There are the description that “ setup model using natural primary teeth placed in denture” and “represent 10 teeth mounted into acrylic blocks”. I can not understand which primary dentition model were used for which experiments.

・For simulation of children’s mouthing behavior, does the setting of the frequency of chewing and the bite pressure unify? The authors used direct hand pressure (L68), was it replicable?

・How many models of primary dentition were used for experiments? If the authors used same model for 5 materials, the error might occur between materials because of abrasion of primary dentition.

Result

・Graph 1 is not need to describe using heat mapping. Describing in tables are simple and suffice.

・Please explain which groups were compared, which kind of representative value were used, and which the statistical analysis methods were used (L147-156).

・I can not understand Table 3. What is the dependent value? Predictors are different from described in text.

Discussion

・The discussion is not written structurally. The authors does not discuss about the result of this experiments. I think the authors should discuss about a comparison among 5 materials. For example, the authors resulted the L/W ratio of silicon was significantly higher than other materials (L138), and what does the result mean? Or, do the authors find value of the simulation reproducing children’s mouthing behaviors?

Minor comments:

Material and Methods

・The same sentences were repeated (L61-69 and L69-76). It may be minor mistake.

Result

・There are no legends in Figure 5. I can not understand what a~d describe.

Result

・Please clarify the unit of measures in Table 1.

・In Figure 6, the left-description is too small to be legible. I recommend delete the description because it is similar to Table 1. The right-description need more detailed explanation, I can not understand what vertical and horizontal axis describe.

・I con not clarify N and Y in Graph 2.

・”(2010)” in L190 is not need.

Round 2

Reviewer 1 Report

The manuscript is acceptable.

Reviewer 2 Report

I think the manuscript was well improved.

I have no further opinions about the manuscript.